# Influence of the Graphene Oxide on the Pore-Throat Connection of Cement Waste Rock Backfill

**DOI:** 10.3390/ma16144953

**Published:** 2023-07-11

**Authors:** Zhangjianing Cheng, Junying Wang, Junxiang Hu, Shuaijie Lu, Yuan Gao, Jun Zhang, Siyao Wang

**Affiliations:** 1School of Transportation and Civil Engineering, Nantong University, Nantong 226019, China; czjn@tongji.edu.cn (Z.C.); 2133110305@stamil.ntu.edu.cn (J.W.); 2233110392@stmail.ntu.edu.cn (J.H.); lushuaijie@stmail.ntu.edu.cn (S.L.); y.gao@ntu.edu.cn (Y.G.); 2College of Civil Engineering, Tongji University, Shanghai 200092, China; 3Nantong Key Laboratory of Intelligent Civil Engineering and Digital Construction, Nantong University, Nantong 226019, China; 4Nantong Taisheng Blue Lsland Offshore Co., Ltd., Nantong 226200, China

**Keywords:** cement waste rock backfill, nano-modification, pore-throat connection, metal intrusion, equivalent pore size distribution

## Abstract

The pore-throat characteristics significantly affect the consolidated properties, such as the mechanical and permeability-related performance of the cementitious composites. By virtue of the nucleation and pore-infilling effects, graphene oxide (GO) has been proven as a great additive in reinforcing cement-based materials. However, the quantitative characterization reports of GO on the pore-throat connection are limited. This study applied advanced metal intrusion and backscattered electron (BSE) microscopy scanning technology to investigate the pore-throat connection characteristics of the cement waste rock backfill (CWRB) specimens before and after GO modification. The results show that the microscopic pore structure of CWRB is significantly improved by the GO nanosheets, manifested by a decrease in the total porosity up to 31.2%. With the assistance of the GO, the transfer among internal pores is from large equivalent pore size distribution to small equivalent pore size distribution. The fitting relationship between strength enhancement and pore reinforcement efficiency under different pore-throat characteristics reveals that the 1.70 μm pore-throat owns the highest correlation in the CWRB specimens, implying apply GO nanosheets to optimizing the pore-throat under this interval is most efficient. Overall, this research broadens our understanding of the pore-throat connection characteristics of CWRB and stimulates the potential application of GO in enhancing the mechanical properties and microstructure of CWRB.

## 1. Introduction

Employing the cemented waste rock backfill (CWRB, which is composed of waste rock, cementitious materials, and water) to fill mining can reduce surface subsidence [1], control rock strata movement [2], reduce water inrush disaster [3] and groundwater resource loss caused by overlying rock failure [4]. Additionally, the CWRB also promotes waste source reduction and resource recycling [5], thereby solving land resource waste caused by waste rocks, additional economic burdens, and potential security threats [3]. The reinforcing effects of the CWRB on the rock movement mainly depend on the mechanical properties of the backfill [6]. If the mechanical parameters of the CWRB cannot meet the expected requirements of the projects, it will cause a series of issues influencing mining safety [7]. Therefore, in recent decades, many scholars devoted themselves to strengthening the mechanical behavior of the CWRB [8,9,10].

It is generally believed that the key to influencing the mechanical properties of cemented backfill materials is the hydration products constituting its bearing structure [11]. Some researchers studied the direction of optimizing the hydration process or products to reinforce the compressive strength of the CWRB [12]. GO is a derivative of graphene, generally obtained by oxidation of natural graphite powder with various oxidants in acidic media [13], which can be viewed as a layer of graphene grafted with oxygen functional groups [14]. Moreover, the thickness of single-layer GO can be several nanometers, whereas Young’s modulus is up to 1 TPa [15], the intrinsic strength is 130 GPa [15], and the specific surface area is about 700–1500 m^2^/g [16]. Recently, by virtue of its great mechanical properties, abundant oxygen-containing functional groups, ultrahigh specific surface area, and ultralow-containing cementitious composites, graphene oxide (GO) have become a special CWRB-enhanced additive [17,18]. Gao et al. [3] reported that applying the industrial GO combined with fly ash could replace 20 wt% cement in CWRB, producing high-performance mine backfill materials. Compared with the plain CWRB, the compressive strength and impermeability properties of the GO-fly ash-modified specimens improved by 5.1–16.9% and 32.7–38.9% by only mixing approximately 0.007 wt% GO [3]. In addition, due to the reduction of cement consumption and utilization of solid waste, this special GO-enhanced CWRB is considered to contribute to environmental protection. The industrial GO modification approach also opens a new pathway for more cost-effective backfill materials. By the way, there are also many modification strategies that can effectively optimize the pore structure and improve the durability of cementitious composites. The direct and effective way is to reduce the water-cement ratio. Using more cement means higher strength and a denser matrix microstructure, which also leads to high costs and CO_2_ emissions. In addition, adding the mineral particles, such as silica fume [19], lime powder [20], and bentonite [21], to cementitious composite materials can improve their mechanical performance and ion penetration resistance. However, these modifications also have certain disadvantages, such as a complex preparation process, high cost, and fluidity impact. In view of the good prospects of the GO-modified CWRB, it is significant to better understand the reinforcing mechanisms of the GO on CWRB for further optimization of the materials.

GO has such great enhancement effects on cementitious composites mainly due to two benefits the nucleation effects [22,23] and pore infilling effects [24,25]. Due to GO owning the abundant oxygen-containing functional groups, the growth of hydration products in the cement matrix could be significantly promoted in the hydration process, resulting in more hydration products being formed in the hardened matrix and reinforcing the pore structure of the cementitious composites [26,27]. Additionally, in the hardened cement matrix, the GO nanosheets would further act as a “wall-like” role to assist the hydration products in optimizing the cement matrixes’ microstructure [28]. Benefited from these two reinforcing mechanisms, GO could reinforce the pore structure of CWRB specimens in two aspects: (I) microstructural refinement of the interfacial transition zone (ITZ) and cement matrix to a denser status and (II) forms barriers at the pore throat junctions to decrease the connectivity among pores [3]. However, in the previous research [29], these mechanisms were mainly analyzed from a qualitative point of view. With the assistance of a new technology called metal intrusion, Gao et al. [29] analyzed the optimization degree of the pore structure of CWRB by GO, and established the relationship between the optimization of pore structure and the improvement of mechanical properties. However, there are few studies on the quantitative characterization of CWRB pore-throat connection and its relationship with mechanical properties.

The degree of cement pore-throat size correlation is an important property affecting the cementitious composites’ mechanical and permeability-related properties [30]. Wardlaw et al. [31] revealed that the pore-throat structure consisted of pores of different sizes arranged randomly in a network and connected by throats, all of which were smaller than the smallest pore. Therefore, pore throat size is correlated, and characterizing the pore-throat of the CWRB is very important but challenging. Hence, in this study, we investigated the pore-throat connection characteristics of the CWRB specimens before and after GO modification with the help of metal intrusion and backscatter electron (BSE) microscope scanning technologies based on the author’s previous research. Firstly, the plain CWRB specimens and the GO-fly ash-modified CWRB specimens were prepared. After that, the 28-day compressive strength and the pore structure of the corresponding age of the CWRB specimens were measured. Then, the effects of the GO nanosheets on the different levels of pore throats in the cement matrix of the CWRB were discussed in detail. Finally, the relationship between different pore-throat modifications and mechanical properties enhancement was further analyzed.

## 2. Methods

### 2.1. Sample Preparation

The preparation processes of the CWRB specimens and the related specimen preparation diagram can be found in the authors’ previous study [29]. Firstly, the industrial GO mixed with the polycarboxylate superplasticizer powders were dissolved in distilled water and dispersed by ultrasonication treatment. The ultrasonication treatment was performed using a VCX-500 W ultrasonicator with a 13 cm probe. The whole ultrasonication process was treated in an ice-water environment to ensure the GO suspension would not be overheated and affect the properties of the GO. The ultrasonic mode was selected as the 3 s pulse and 3 s pause mode with a 150 W ultrasonication power and 10 min ultrasonication time [32]. The mass ratio of the industrial GO and polycarboxylate superplasticizer powders were chosen as 0.08 wt% and 0.64 wt% of the suspension. A previous study [33] suggested that this mixing ratio would trade off the reinforcing effects and the economic benefits of the GO nanosheets. A reference group (Ref-group), with the same mixed proportion of the PC without GO, was used to better investigate the effects of GO on the pore-throat connection and the corresponding properties of the CWRB.

After the industrial GO dispersion, for the GO-reinforced group, the ordinary Portland cement (OPC, type P.O. 42.5) and fly ash powders were mixed together under the mass ratio of 4:1 with a 5 min dry stirring to avoid the uneven distribution in the hardened CWRB specimens. After that, the prepared GO suspension was poured into the powders’ mixture with a 0.6 water-to-cement (W/C) ratio to generate the cementitious slurry. The Ref-group suspension was also mixed with the plain OPC powder under the same W/C ratio. The detailed mix design of the GO- and Ref-group is shown in Table 1. Afterward, the fresh cementitious slurry was poured into the coal gangue. The coal gangue, with a density of 2.25 g/cm^3^, was employed as the rock backfill aggregates in this work. Then, the fresh CWRB mixtures were poured into the cylindrical mold with the size of 50 × 100 mm^2^ for the preparation of the compressive mechanical properties tests.

According to the ASTM C192/C192M-20a [34], the diameter of the test specimens must be at least three times higher than the biggest aggregate particle to avoid the side effects affecting the mechanical behavior. Therefore, in this study, the coal gangues were sieved into seven levels followed by the previous report [35] as 0–0.5 mm, 0.5–1.0 mm, 1.0–1.5 mm, 1.5–2.5 mm, 2.5–5 mm, 5–8 mm, and 8–10 mm, respectively. The Talbot gradation theory [36] was then applied to partition the gradation of the backfill aggregates. Four Talbot indices, 0.2, 0.4, 0.6, and 0.8, were adopted in CWRB specimen preparation.

After the corresponding weight of the coal gangue aggregates was selected, the gangues were then mixed with prepared fresh cementitious slurry with a concentration ratio of 1:0.35 using a concrete mixer and stirred for 5 min, and poured into a 50 × 100 mm^2^ under another 3 min vibration to ensure the compactness of the CWRB. The specimens, after pouring them, were placed in a concrete curing box for 28 days of curing with a temperature and humidity of 20 ± 1 °C and 95%, respectively.

### 2.2. Measurement

To investigate the influence of the graphene oxide on the pore-throat connection of cement waste rock backfill, the CWRB specimens, after 28 days of curing, were soaked in an ethanol solution to avoid the CWRB continuing hydration. Afterward, the cylindrical specimens were cut into small cubes with the size of 5 × 5 × 5 mm^3^. After that, the cut CWRB cubes were placed in a ventilated oven for 24 h at 105 ± 2 °C for drying. Then, the metal intrusion method was applied to fill the CWRB specimens with Field’s metal. The purpose of the metal intrusion is to let the pore structure in CWRB could be better displayed under a scanning electron microscope (SEM). The specific operation steps and principles of metal intrusion follow the literature [37]. After metal intrusion, the cubes were soaked in the epoxy impregnation for the preparation of the SEM characterization. Before scanning electron microscope tests, the specimens were first polished. The polishing was divided into two main steps by using six grades of coarse sandpaper (178 μm, 61 μm, 38 μm, 23 μm, 13 μm, and 5.5 μm) and four fine grades of diamond grit (2.5 μm, 1 μm, 0.5 μm, and 0.25 μm), respectively [36].

The polished CWRB specimens were then characterized using a high-resolution field emission SEM with a backscattered electron (BSE) detector. The dwell time and scanning electron for the BSE characterization were selected as ten µs and five keV, respectively. To alleviate the influence of electrical charges during the imaging process, one two-nanometer-thickness gold film was deposited on the surface of polished specimens. The magnification of each BSE image is set to 320 times, with 1280 × 960 pixels square. More details concerning the metal intrusion and BSE image characterization have followed the literature [37]. By the way, three specimens for each group (four Ref-group and four GO-group) were tested in BSE image processing, and 8 BSE images were taken for each specimen. After that, the cylindrical CWRB specimens at 28-day age were used for the compressive strength tests according to the authors’ previous reports [29]. The mechanical test results are listed in the literature [29].

## 3. Results and Discussion

### 3.1. Pore-Throat Characterization

Considering the pore-throat connection of the CWRB significantly influence the related mechanical performance, permeability-related properties, and durability of the hardened specimens. The modified mechanisms of the graphene oxide nanosheets on the micro/nanopore structure of the CWRB were first investigated. Figure 1a shows one typical BSE image of the CWRB specimens after metal intrusion treatment. A previous study suggested that the cementitious composites belong to a porous polyphase structure, the chemical composition of each phase is different, the BSE electron image can be used to distinguish different phases, the cement-based materials phase qualitative and quantitative analysis [38]. The image magnification is 320 times with 1280 × 960 pixels. In order to better distinguish the different CWRB terms in the BSE image, especially the substrate materials and the pores, the image was first treated with binarization. The binarization of the image is conducive to the further processing of the image, making the CWRB images simple. The amount of data was reduced, which could highlight the outline of the objects in the cementitious matrix. The IsoData thresholding method was used to calculate the optimal threshold, referring to the literature [39]. The image after binarization treatment is exhibited in Figure 1b, where black represents the cement matrix and white represents the pores in CWRB specimens. Afterward, the pore structure, pore size distribution, and pore-throat connection of the CWRB specimens could be well investigated.

In order to more intuitively display the pore-structure characteristics such as pore size, distribution, and pore-throat connection in the BSE image, the pores of different sizes in the CWRB specimens are further distinguished by color mapping, and the equivalent diameter (representing the corresponding diameter of the circle which has the same surface area of pores [40]) was used to characterize the pore size. As shown in Figure 1c, the original white pores are filled with purple and red, where the purple part represents pores with an equivalent diameter close to or much larger than 100 μm, while the red part represents pores with an equivalent diameter of 0–14.3 μm. In addition, due to the existence of pore-throats, the original small pores are interconnected by throats and become larger pores, which greatly changes the pore structure of specimens.

Four kinds of pore throats with equivalent diameters (0.34, 1.02, 1.70, and 2.38 μm) were defined to investigate the effect of pore-throat size on the pore structure of the specimen and the effect of GO on the pore-throat and the corresponding pore structure. Taking a pore-throat with the size of 1 pixel (0.34 μm) as an example, we consider that the pores are independent of each other when the distance between pores (pore-throat size) is less than 0.34 μm, and the pores are connected when the distance between pores (pore-throat size) is larger than 0.34 μm. The equivalent diameter of the smallest pore-throat is 0.34 μm (the actual size corresponding to a single pixel), which is calculated from the pixel size (1280 × 960 pixels square) and the actual size (435.2 × 326.4 microns square) of the BSE image.

The equivalent pore size distribution of CWRB specimens under different pore throat characteristics was presented in Figure 1c–f. As the equivalent diameter of pore-throat increases, the equivalent diameter of pores in specimens decreases significantly, and a small part of the original 100 μm pores (marked in purple) are firstly divided into 15 μm pores (marked in orange) and 30 μm pores (marked in green). Subsequently, most of the 100 μm pores (marked in purple) are divided into 10 μm pores (marked in red), 20 μm pores (marked in yellow), and 60 μm pores (marked in light blue). At the same time, 30 μm pores (marked in green) turned into pores marked in yellow and red, and 15 μm pores (marked in orange) almost all turned red. When the equivalent diameter of the pore-throat is 2.38 μm, the BSE image of specimens is entirely covered by pores with an equivalent pore diameter of 0–20 μm. When the equivalent diameter of the pore-throat is 2.38 μm, the BSE image of specimens is entirely covered by pores (marked in red and orange) with an equivalent pore diameter of 0–15 μm.

### 3.2. Effect of GO on Pore-Throat of CWRB Specimens

To further reveal microstructural alteration of CWRB specimens caused by GO, the changes in pore size, equivalent pore size distribution, and porosity of samples with different Talbot gradation indexes before and after adding GO were compared. As shown in Figure 2a, the BSE image of the Ref-group specimen presents a single interpenetrating pore accompanied by partially separated pores with low equivalent diameters under the pore-throat equaling 0.34 μm. An apparent pore-splitting phenomenon appears after the addition of GO, especially at the Talbot gradation index n = 0.2 (i.e., Group-1), the through pores are divided into many pores of different sizes, and the ability of GO to segment pores gradually weakens with the increasing Talbot gradation index. Figure 2b shows the equivalent pore size distribution of Ref- and GO-enhanced CWRB specimens under 0.34 μm pore-throat characterization. The pores in the Ref-1 specimen are mainly concentrated in the range of 100–250 μm, the pores in Ref-2 and Ref-4 specimens are mainly concentrated in the range of 200–250 μm while those of Ref-3 specimen are in the range of 150–200 μm. With the addition of GO, 100–250 μm pores in the Ref-1 specimen were transformed into a large number of 0–100 μm pores, and some of the 200–250 μm pores in the Ref-2 specimen were transformed into 150–200 μm pores. Compared with GO-1 and GO-2 specimens, the pore distribution in GO-3 and GO-4 did not change significantly, which confirms that GO has a limited effect on optimizing the microstructure of specimens with high Talbot gradation index.

The relationship between cumulative porosity and equivalent diameters of pores in the matrix is shown in Figure 2c. The first half of most curves are flat, indicating that there are few pores with equivalent diameters corresponding to those ranges, while the latter half of the curve exhibits an apparent plunge, indicating that most of the pores are large pores, and most of them are distributed in the range of 200–250 μm, which is consistent with the conclusion that the pores in CWRB specimens are mostly large pores with a single penetration, as concluded in Figure 2a,b. Additionally, the cumulative percentage of the porosity in the CWRB matrix with different aggregate Talbot indices under 0.34 μm pore-throat characterization is shown in Figure 4d. It can be found that with a certain amount of GO incorporation, the porosity of the CWRB specimen with aggregate Talbot index n = 0.6 (i.e., Group-3) remains unchanged. For the other aggregate Talbot indices, the porosity decreases to some extent, ranging from 6.0% to 21.1% (reaching the maximum at Group-2). The above phenomena indicate that GO effectively optimizes the microstructure of CWRB specimens, and as previous studies have concluded, GO significantly changes the pore structure of CWRB specimens through the filling and nucleation effect. On the one hand, due to the tiny physical size of GO, it can effectively fill the micro-pores in cement consolidation, thereby optimizing the microstructure of specimens (filling effect). On the other hand, GO has an ultra-high specific surface area and abundant oxygen-containing functional groups, which can provide additional nucleation sites for the early hydration of cement, significantly promote the hydration reaction of cement, and make the hydration products grow along the surface of GO, and act together with the produced hydration products in the cement matrix (nucleation effect). The dense hydration products that grow on the surface of GO will gradually develop into a unique wall-like structure, which promotes the division of large pores into smaller ones in the cement matrix.

When the pore-throat size is 3 pixels (which is equal to 1.02 μm), as shown in Figure 3a,b, different from the single connected large pores in Figure 2a, under the lower Talbot gradation index (n = 0.2), many separation pores with different sizes appear in the BSE images of the Ref-group specimens, and the equivalent diameter of these pores is almost less than 100 μm. With the increase of aggregate Talbot index, on the one hand, the BSE image is gradually covered by pores with an equivalent diameter more prominent than 100 μm and a small number of pores with an equivalent pore diameter smaller than 20 μm. On the other hand, the number of pores with large equivalent diameters increases significantly. The pore structure of the CWRB specimens changed significantly after adding GO, where the pore distribution characteristics shifted from the sizeable equivalent pore size distribution to the small equivalent pore size distribution. The appearance of smaller equivalent diameter pores in the BSE images corroborates this perspective. Moreover, as represented in Figure 3c,d, GO has the best optimization effect on the porosity of the Group-1 specimen, with an optimization range of 21.1%, followed by Group-4, and there is no noticeable optimization effect in Group-3, while for Group-4 has a deteriorating effect.

When the pore-throat size is 5 pixels (which is equal to 1.70 μm), as shown in Figure 4a, only a small number of large pores (equivalent diameter is 60–80 μm) are present in BSE images of Ref-4 specimen, and the BSE images in the other Ref-Groups are covered by a red and orange pore (equivalent diameter are 0–20 μm) and a few yellow and green pores (equivalent diameter are 2–40 μm). Similar to Ref-Group, the BSE images of GO-Group specimens (excluding GO-4) are almost entirely covered by red and orange pores. In contrast, a small number of blue pores (equivalent diameter ~60 μm) are present in GO-4 specimens. On the other hand, unlike Figure 2a and Figure 3a, when the pore-throat size is 5 pixels (which is equal to 1.70 μm), in the BSE image of CWRB specimens, there are a large number of tiny pores (equivalent diameter is 20–40 μm) rather than connected large pores (equivalent diameter is 60–1000 μm). This significant change is due to the presence of a large number of pore throats (greater than 1.02 μm and less than 1.70 μm) in CWRB specimens. As the size of pore-throat increases to 1.70 μm, the large number of 1.02–1.70 μm pore-throats that act as connecting pores are no longer recognized as pore channels, resulting in a large number of large pores being split into tiny pores.

Moreover, comparing the BSE images of Ref-Group and GO-Group, it is easy to find that the incorporation of GO leads to a decrease in the number of pores with large equivalent diameters and an increase in the number of pores with small equivalent diameters. The results in Figure 4b correspond to the BSE image, which can be clearly observed that the addition of GO leads to an abnormally significant decrease in the mercury intrusion into pores with large equivalent diameters, and the pore distribution characteristics are significantly transferred from large equivalent pore size distribution to small equivalent pore size distribution. Furthermore, the results of porosity and pore distribution in Figure 4c,d also indicate that, no matter which Talbot index is, GO is always effective in optimizing the microscopic pore structure of CWRB specimens, and the enhancement efficiency is 4~26.9%. Therefore, based on the above reasons, it is considered that GO has the gradely enhancement effect on the microscopic pore-structure of CWRB specimens when the pore-throat is equal to 1.70 μm.

When the pore-throat size is 7 pixels (which is equal to 2.38 μm), most areas in the BSE image of the Ref-Group specimens are covered by red pores with an equivalent diameter of less than 10 μm and small areas were covered by orange pores (equivalent diameter are 0–20 μm). However, the BSE image of the Ref-4 sample presents partial yellow and green pores with an equivalent diameter of 20–40 μm, as shown in Figure 5a. With the incorporation of GO, only red and orange pores remained in the BSE image of all CWRB specimens. Unlike Figure 4b, the equivalent pore size distribution characteristics of CWRB specimens (except for Group-4) with different Talbot indices in Figure 5b are similar and do not alter due to GO blending. In addition, comparing the porosity changes of CWRB specimens before and after GO modification in Figure 2c and Figure 5c and Figure 2d and Figure 5d, it can be found that GO has the best enhancement effect on the microstructure of the CWRB specimens under pore-throat equaling 2.38 μm with reinforcing ratio varying from 7.4% to 31.2% and better enhancement effect on large pore-throat (equaling 1.70 and 2.38 μm) than small pore-throat (equaling 1.70 and 2.38 μm).

### 3.3. Effect of GO on the Number of Pores with Different Equivalent Pore Sizes of CWRB Specimens

To further investigate the influence of GO on the number of pores with different equivalent pore sizes under different pore-throat characterization, the change ratio of pores with different equivalent pore sizes was calculated and counted as shown in Figure 6, where a change ratio greater than 0 indicates an increase and less than 0 indicates a decrease in pore number. Figure 6a shows the change in pores number with different equivalent pore sizes when pore-throat size is equal to 0.34 μm. It is clear that GO has the best improvement in CWRB specimens for Group 1. Its 150–200 μm and 200–250 μm pores completely disappear (the change ratio is −100%) and resulted in a significant increase in the number of pores in the range of 0–150 μm, especially in the range of 50–100 μm by a staggering 2130%. At the same time, that also means that the pores in CWRB specimens were segmented into smaller pores caused by GO. Compared with Group 1, the effect of GO on pore structure improvement for Group 2 was weakened. The pore number of 200–250 μm pores decreased by 44.7%, while the pore number of 50–100, 100–150, and 150–200 μm pores increased by 130.8%, 474.3%, and 563.3%, respectively.

Similar to Group 2, the number of 200–250 μm pores in Group 4 decreased slightly, and the number of 0–50 μm pores increased slightly, different from 0 to 50 μm pores in Group 2, which remained constant. However, GO caused the deterioration of the microscopic pore structure of Group 4, manifested by the decrease of 50–200 μm pores and the increase of 200–250 μm pores. When the pore throat is equal to 1.02 μm (result as represented in Figure 5b), GO can enhance the microstructure of all CWRB specimens to a certain extent, as evidenced by the significant reduction in the number of large pores and the significant increase in the number of pores with relatively small equivalent diameters. As the pore throat increases to 1.70 μm (as shown in Figure 5c), except for Group 4, the 100–200 μm pores in other CWRB samples nearly disappear, which is the reason for the change rate of 0. Notably, the number of pores with all equivalent diameters in the Group 1 sample decreases, which means that the total porosity decreases. The conclusions of the other groups are consistent with that pore-throat is equal to 1.02 μm. In addition, when the pore-throat is equal to 2.38 μm, the pore-structure of CWRB specimens still exhibits a decrease of large pores and an increase of tiny pores, which is different from the pore-throat equal to 0.34 μm, 1.02 μm, and 1.70 μm, with a decrease in all pores more significant than 50 μm and an increase in pores smaller than 50 μm (Group 1 shows the decrease in 0–50 μm pores).

### 3.4. Relationship between Pore-Throat Modifications and Mechanical Properties of CWRB Specimens

In order to clarify the interaction law of compressive strength and porosity of CWRB specimens before and after GO modification under different aggregate Talbot indices, the mechanical test results of CWRB specimens in the previous study [22] (as shown in Figure 7a) were used to establish the fitting relationship between compressive strength enhancement efficiency and the Porosity enhancement efficiency under different pore-throat characterization (as shown in Figure 7b). The fitting coefficient is the highest at 0.987 under pore-throat equaling 1.70 μm. Therefore, we consider that 1.70 μm pore-throat characterization can accurately describe the microscopic pore-structure of the CWRB specimens, and recommend this parameter to be adopted in future studies.

## 4. Conclusions

In this study, advanced metal intrusion and backscattered electron (BSE) microscopy scanning techniques were applied to investigate the pore-throat connectivity of CWRB specimens before and after GO modification with different pore-throat characteristics and the fitting relationship between the intensity enhancement efficiency and pore enhancement efficiency was established, with following conclusions:(1)GO is able to optimize the microstructure of CWRB specimens under different pore-throat characteristics, and optimization amplitude hits up to approximately 32.1%, which is due to the nucleation effect that promotes the growth of hydration products and the pore-infilling effect that fills and divides the pores and pore-throat, both of which contribute to the optimization of microstructure;(2)With the increase of pore-throat size, the porosity of CWRB specimens showed a decreasing trend, and the pores shifted from large equivalent pore size distribution to small equivalent pore size distribution, which was due to the fact that the original small pore throat was no longer used as a channel to connect the pores, but split the original large pores;(3)A large number of pore-throat with equivalent diameters of 1.02–1.70 μm existed in the CWRB specimens, and GO significantly reduced the number of pore-throats in this size, effectively partitioning the pores and improving the microscopic pore-structure of CWRB;(4)The 1.70 μm pore-throat characteristic can reflect the microscopic pore-structure of CWRB specimens more accurately than others, which is recommended for future studies.

## Figures and Tables

**Figure 1 materials-16-04953-f001:**
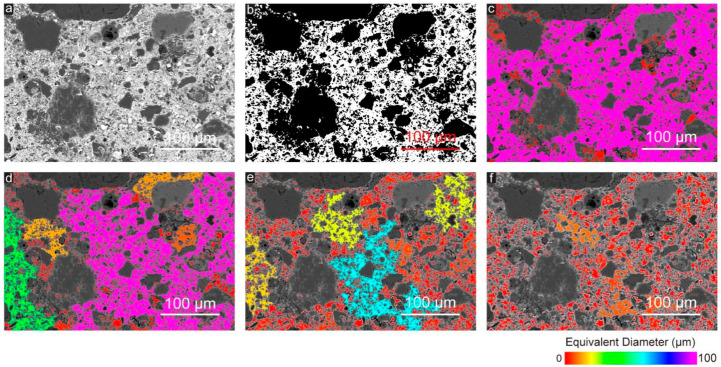
(**a**) One BSE image of the cement matrix area of the CWRB specimens; (**b**) The BSE image after binarization treatment. Equivalent pore size distribution of the imaged samples with different pore-throat characterization; (**c**) 0.34 μm; (**d**) 1.02 μm; (**e**) 1.7 μm; and (**f**) 2.38 μm indicated using a color bar.

**Figure 2 materials-16-04953-f002:**
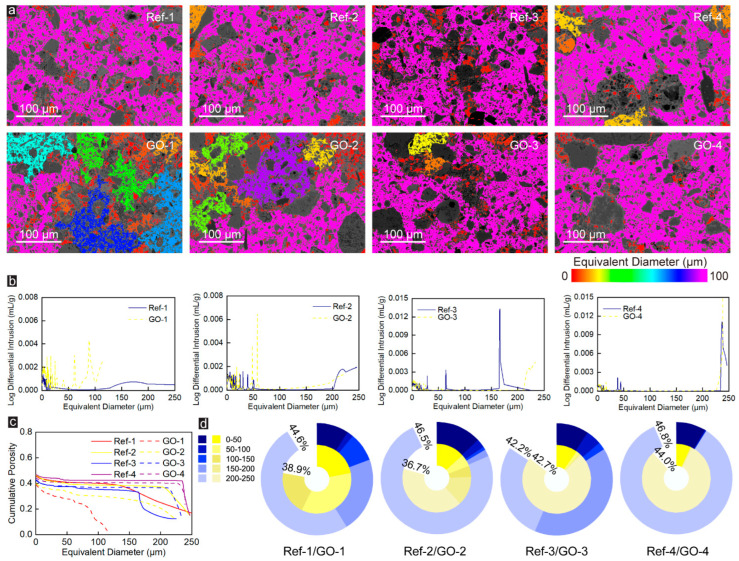
(**a**) The pore sizes are indicated by the colormaps under the pore-throat equaling 0.34 μm. (**b**) The corresponding equivalent pore size distribution of the Ref- and GO-reinforced CWRB specimens with different aggregate Talbot indices under 0.34 μm pore-throat characterization. (**c**) The cumulative porosity versus the characterized equivalent pore diameter in the tested cement matrix. (**d**) The cumulative percentage of the porosity in the CWRB matrix with different aggregate Talbot indices under 0.34 μm pore-throat characterization.

**Figure 3 materials-16-04953-f003:**
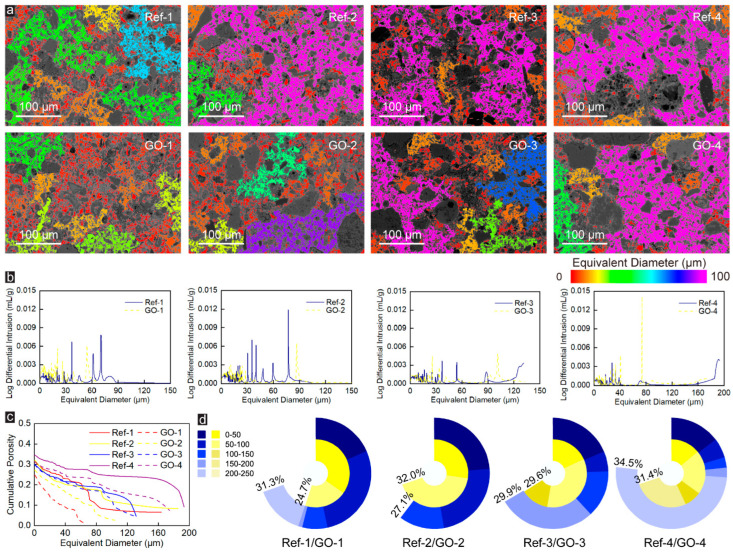
(**a**) The pore sizes are indicated by the colormaps under the pore-throat equaling 1.02 μm. (**b**) The corresponding equivalent pore size distribution of the Ref- and GO-reinforced CWRB specimens with different aggregate Talbot indices under 1.02 μm pore-throat characterization. (**c**) The cumulative porosity versus the characterized equivalent pore diameter in the tested cement matrix. (**d**) The cumulative percentage of the porosity in the CWRB matrix with different aggregate Talbot indices under 1.02 μm pore-throat characterization.

**Figure 4 materials-16-04953-f004:**
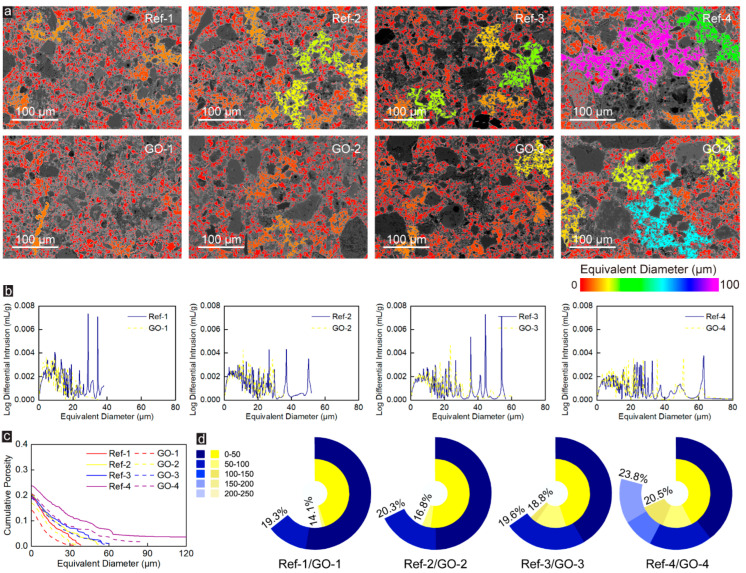
(**a**) The pore sizes are indicated by the colormaps under the pore-throat equaling 1.70 μm. (**b**) The corresponding equivalent pore size distribution of the Ref- and GO-reinforced CWRB specimens with different aggregate Talbot indices under 1.70 μm pore-throat characterization. (**c**) The cumulative porosity versus the characterized equivalent pore diameter in the tested cement matrix. (**d**) The cumulative percentage of the porosity in the CWRB matrix with different aggregate Talbot indices under 1.7 μm pore-throat characterization.

**Figure 5 materials-16-04953-f005:**
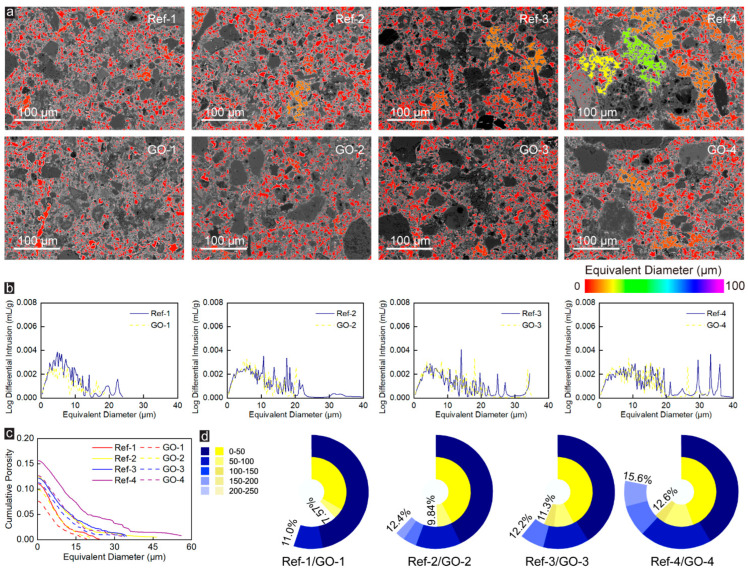
(**a**) The pore sizes are indicated by the colormaps under the pore-throat equaling 2.38 μm. (**b**) The corresponding equivalent pore size distribution of the Ref- and GO-reinforced CWRB specimens with different aggregate Talbot indices under 2.38 μm pore-throat characterization. (**c**) The cumulative porosity versus the characterized equivalent pore diameter in the tested cement matrix. (**d**) The cumulative percentage of the porosity in the CWRB matrix with different aggregate Talbot indices under 2.38 μm pore-throat characterization.

**Figure 6 materials-16-04953-f006:**
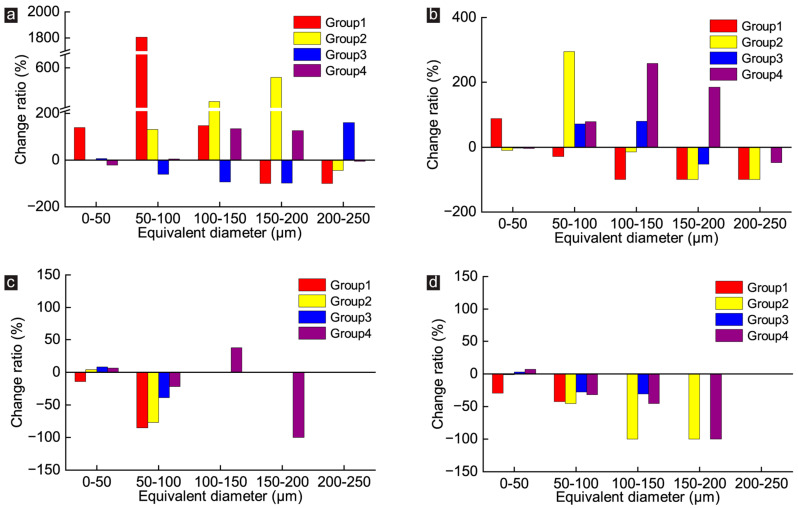
Change ratio of pores with different equivalent pore sizes of CWRB specimens under the pore-throat equaling (**a**) 0.34 μm. (**b**) 1.02 μm. (**c**) 1.70 μm. and (**d**) 2.38 μm.

**Figure 7 materials-16-04953-f007:**
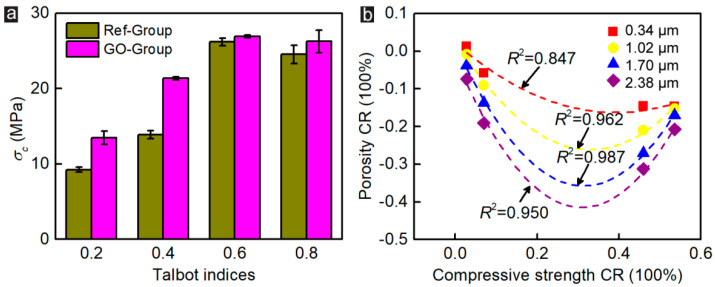
(**a**) The compressive strength of Ref- and GO-reinforced CWRB specimens with different aggregate Talbot indices. (**b**) Fitting relationship between compressive strength CR and porosity CR of CWRB specimens.

**Table 1 materials-16-04953-t001:** Mix proportions of the Ref and GO-FA hybrid modified CWRB specimens [29].

Specimen	W/C	G/s (wt%)	P/s (wt%)	F/SL (wt%)	C/SL (wt%)	C/B (wt%)	G/B (wt%)
Ref-group	0.6	--	0.64	--	62.5	16.2	--
GO-group	0.6	0.08	0.64	12.5	50.0	13.0	0.008

(Note: G/s and P/s represent GO and PC to suspension weight percentages; F/SL and C/SL represent FA and cement powder to mixing cement-based slurry weight percentages; C/B and G/B represent cement and GO to backfill specimens weight percentages).

## Data Availability

The data presented in this study are available on request from the corresponding author.

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
