# Peer review of "Influence of the Graphene Oxide on the Pore-Throat Connection of Cement Waste Rock Backfill"

_materials, 2023, doi:10.3390/ma16144953_

Round 1
Reviewer 1 Report
The paper appears to be well-written and includes detailed descriptions of the methods and results. However I have a few remarks.
1. I noticed that the study does not provide explicit information about the number of samples used in each group, namely the Ref group and the GO group. Without knowing the exact number of samples, there is a possibility that the results could be influenced by outliers or random variation, especially if only a few samples were tested.It would be beneficial for the study to clarify the number of samples used in each experimental condition to ensure the statistical robustness of the conclusions.
2. The study was conducted under very specific conditions, including ultrasonic time and power, a specific water-to-cement ratio, and a controlled curing environment. Still, it's important to consider the generalizability of the results to real-world scenarios. To address this concern, I suggest that the study investigate the effects of GO under a wider range of conditions to ensure the applicability of the results to practical situations. This would help strengthen the overall validity of the study and increase its practical relevance.
3. The threshold used for binarization is not explicitly stated by the authors. I am aware of the importance of the threshold parameter in binarization and its potential impact on the results. While it is possible that the authors used a standard method such as Otsu's method, which calculates the optimal threshold based on the histogram of the image, this is only a conjecture in the absence of explicit information in the paper, and in my opinion some explanation should appear in the paper.
Author Response
Please see the attactment.

Reviewer 2 Report
Quite interesting study about metal intrusion and BSE to investigate the pore-throat connection characteristics of the cement waste rock backfill specimens before and after GO modification. The results are sound and clear and the experiences seem well planned
Author Response
Please see the attactment.

Reviewer 3 Report
Dear Authors,
I have only a few comments, since structure and content of the article are very thoroughly elaborated:
- Abstract/Introduction: give a definition what CWRB is. Should it not be "cemented waste rock backfill" (cemented in the sense of bond by cement and not waste made of cement); give broader basic information about GO!
- Give a short information about how GO is produced. You write that it´s environment-friendly: please provide evidence for this statement! Is GO environmentally friendly, what´s its ecological footprint?
- What other methods would there be to reduce the pore space or permeability in this material? Advantages/disadvantages?
- Methods: a Picture of the sample production / manufacture of specimens would be helpful! Also from the equipment used (sample during examination), e.g. sample under SEM.
- Fig. 1: Explain in more detail what can be seen, what do different colours display, whats happening between 1c and 1f? (attention numbering is wrong) This would be also helpful to interpret the following illustrations.
- expressions like "excellent" and "superior" does not sound scientific, please change
- Chapter 2.3 is very long - may subdivide
Many thanks and best regards
English is fine, improve some expressions.
Author Response
Please see the attactment.
